# Tick-Borne Flaviviruses Depress AKT Activity during Acute Infection by Modulating AKT1/2

**DOI:** 10.3390/v12101059

**Published:** 2020-09-23

**Authors:** Joshua M. Kirsch, Luwanika Mlera, Danielle K. Offerdahl, Marthe VanSickle, Marshall E. Bloom

**Affiliations:** Biology of Vector-Borne Viruses Section, Laboratory of Virology, Rocky Mountain Laboratories, NIAID/NIH, Hamilton, MT 59840, USA; maoy12@gmail.com (J.M.K.); luwanikamlera@arizona.edu (L.M.); offerdahld@niaid.nih.gov (D.K.O.); marthe.vansickle@gmail.com (M.V.)

**Keywords:** Langat virus, tick-borne flavivirus, AKT, apoptosis, viral persistence

## Abstract

Tick-borne flaviviruses (TBFVs) are reemerging public health threats. To develop therapeutics against these pathogens, increased understanding of their interactions with the mammalian host is required. The PI3K-AKT pathway has been implicated in TBFV persistence, but its role during acute virus infection remains poorly understood. Previously, we showed that Langat virus (LGTV)-infected HEK 293T cells undergo a lytic crisis with a few surviving cells that become persistently infected. We also observed that *AKT2* mRNA is upregulated in cells persistently infected with TBFV. Here, we investigated the virus-induced effects on AKT expression over the course of acute LGTV infection and found that total phosphorylated AKT (pAKT), AKT1, and AKT2 decrease over time, but AKT3 increases dramatically. Furthermore, cells lacking AKT1 or AKT2 were more resistant to LGTV-induced cell death than wild-type cells because they expressed higher levels of pAKT and antiapoptotic proteins, such as XIAP and survivin. The differential modulation of AKT by LGTV may be a mechanism by which viral persistence is initiated, and our results demonstrate a complicated manipulation of host pathways by TBFVs.

## 1. Introduction

Tick-borne flaviviruses (TBFVs) are members of the *Flaviviridae* family, which comprises more than 70 globally distributed single-stranded (+)RNA viruses [1]. The ss(+) RNA genome measures ~11 kb and contains a single open reading frame, which is translated as a polyprotein and subsequently cleaved by host and viral proteases into 10 different proteins [2]. The individual proteins are 3 structural proteins: C (capsid), M (membrane), and E (envelope) and 7 nonstructural proteins: NS1, NS2A, NS2B, NS3, NS4A, NS4B, and NS5. NS5 is the RNA-dependent viral RNA polymerase and it also antagonizes host cells IFN responses [3,4]. NS2B and NS3 form a protease complex that cleaves sections of the viral polyprotein [5]. The functions of the remaining nonstructural proteins are not completely known, but NS1, NS2A, NS4A, and NS4B are implicated in virus replication and/or evasion of innate immune signaling [6,7,8,9,10,11]. The genome also contains 5′ and 3′ untranslated regions that flank the open reading frame and are important for virulence and viral function [12,13,14,15].

Many flaviviruses infect humans using an arthropod vector, such as a mosquito or tick [16]. The TBFV family includes tick-borne encephalitic virus (TBEV), Powassan virus (POWV), and Langat virus (LGTV). TBEV is endemic to Europe and Asia. POWV is known to circulate in eastern Russia while also being the only TBFV found in North America, particularly the northeast states such as New York and New Hampshire, as well as eastern regions of Canada [17]. LGTV is a member of the TBEV serocomplex, is not known to cause disease in humans, and is used as a biosafety level-2 model of TBFVs [12]. This virus was discovered in Malaysian ticks [18]. The closely related mosquito-borne flaviviruses (MBFVs) include Zika virus (ZKV), Dengue virus (DENV), West Nile virus (WNV), and Japanese encephalitis virus (JEV).

TBEV is the most medically important tick-borne flavivirus both in terms of number of cases and fatalities. Despite a widely available vaccine, TBEV morbidity in Europe increased by 400% between 1974 and 2003 [19]. There were over 2500 new cases of TBEV in 2016 in Europe alone [20]. Due to climate change, the TBEV vector range has increased in recent years, as was seen in the Jeseníky mountains in the Czech Republic [21]. Additionally, increased surveillance has recently identified TBEV in the UK and the Netherlands [22,23]. Most TBFV infections present in a biphasic manner. An early acute phase consisting of mild febrile illness [24] is followed by a second phase often characterized by more severe symptoms, such as encephalitis and meningoencephalitis [25]. Patients who survive the encephalitic phase can suffer long-term debilitating neurological sequalae, such as confusion, tremors, ataxia, and cranial nerve paralysis [25]. In the case of TBEV, up to 60% of patients experience both phases of infection, and of this subset of patients, 30% had long-term sequelae [26,27]. The long-term neurological symptoms are thought to be the result of viral persistence or neurological tissue damage acquired during acute infection or a combination of the two. In one interesting case, TBEV was isolated from the brain of a man who was bitten by a tick 10 years prior and subsequently died of progressive neurological illness [28].

During infection and persistence, TBFVs co-opt host signaling pathways for their benefit. Understanding how TBFVs manipulate these pathways might inform new targets for drug discovery [29]. Some of the hijacked pathways include the ATF6/UPR pathway and the PI3K-AKT pathway [30,31]. The ATF6/UPR pathway is activated during TBEV infection and is necessary for productive viral replication [30]. In other viral systems, the PI3K-AKT pathway has been shown to be crucial for a variety of viral functions including entry and host cell survival [32]. For example, studies using Zaire Ebola virus showed that the virus requires PI3K-AKT activation to be released into the cytoplasm after its initial uptake [33]. Human cytomegalovirus, a DNA virus, utilizes the PI3K-AKT pathway to avoid apoptosis through interactions with its major immediate–early proteins, and to promote replication [34,35]. Flaviviruses have also been shown to manipulate the PI3K-AKT pathway. For example, during DENV and JEV infection, the PI3K-AKT pathway is upregulated early in infection to avoid virus-induced apoptosis [36]. In a similar vein, ectopic expression of a variety of MBFV capsid proteins has been found to upregulate AKT signaling [37,38]. Inhibition of PI3K was shown to significantly decrease TBEV replication in dendritic cells in the presence of tick saliva [31]. However, ZIKV NS4A/B have been demonstrated to downregulate active AKT (phosphorylated AKT/pAKT) to induce autophagy, thus priming the cellular environment for virus replication [39].

The PI3K-AKT pathway is activated beginning with PI3K converting PIP2 to PIP3, which allows AKT to bind to PIP3 and undergo a conformational change to be phosphorylated [40]. PI3K is activated by a wide variety of G protein-coupled receptors and their conjugate binders, which include growth factors and antigens [41,42]. Multiple subsequent phosphorylation events occur on AKT residues before the protein reaches its full activity. pAKT is a promiscuous kinase and is reported to have over a hundred different substrates [40]. As such, it has different roles in various cell processes, including translation, apoptosis, glucose metabolism, and immune signaling. AKT has three homologous isoforms: AKT1, AKT2, and AKT3, and each isoform is encoded by a different gene [43]. A growing body of evidence suggests that each AKT isoform has unique functions, both on downstream effectors and regulatory feedbacks on the other isoforms [44,45,46]. AKT1, not AKT2 or AKT3, is a key messenger for IL-3′s antiapoptotic activity in myeloid cells [44]. Mice lacking AKT1 present inhibited growth and increased sensitivity to apoptosis in thymocytes and embryonic fibroblasts [47]. AKT2 influences glucose metabolism, and humans lacking functional AKT2 are predisposed to diabetes [48]. AKT3 knock-out (KO) mice demonstrate impaired neural development, in the absence of impaired glucose metabolism [49].

The PI3K-AKT pathway is yet to be studied in the context of flavivirus persistence despite growing evidence of the pathway’s involvement. In a transcriptomic analysis of cells persistently infected with LGTV, *AKT2* mRNA was upregulated compared to acutely infected cells [50]. In addition, ZIKV persistence in the cerebrospinal fluid of rhesus monkeys was associated with an upregulation of the PI3K-AKT pathway members such as the *mTOR* and *PI3K* genes [51]. The persistence of other viruses, including Epstein–Barr virus and hepatitis C virus, also requires the PI3K-AKT pathway to prolong cell survival by avoiding apoptosis [32].

In this study, we extended our previous work identifying increased *AKT2* mRNA [50] by investigating the role played by AKT isoforms during the acute phase of infection up through the lytic crisis. We found that AKT1 and AKT2 were downregulated at the protein level during the acute phase of infection, but AKT3 was upregulated. Infected cells also displayed decreased amounts of AKT phosphorylated at T308 and S473. *AKT1* mRNA levels also decreased late in the acute phase. In addition, cells deficient in AKT1 or AKT2 were more resistant to LGTV-induced cell death due to higher amounts of pAKT at a late stage in infection. These results demonstrate that LGTV induces apoptosis due to decreases in specific AKT isoforms, and this decrease requires the expression of both AKT1 and AKT2. Our results show a novel mechanism for viral-induced cell death through depletion of AKT1 and AKT2.

## 2. Materials and Methods

### 2.1. Cells

Human embryonic kidney (HEK) 293T/I7 cells (ATCC CRL-11268, Manassas, VA, USA) were maintained in complete Dulbecco’s MEM (DMEM) containing 10% FBS and Anti-Anti (Gibco, Life Technologies, Carlsbad, CA, USA).

### 2.2. Virus Infections and Establishment of Persistently Infected Cultures

Infections were performed using Langat TP21 virus at a multiplicity of infection (MOI) of 0.1. The virus was generated from transfecting Vero cells (ATCC CCL-81, Manassas, VA, USA) with full length viral RNA transcribed from a cDNA clone (GenBank ID EU790644) as described in [52]. Following rescue, the virus was propagated in Vero cells and titrated on the same cells using an immunofocus assay as previously described [53]. Cells were incubated with the infecting inoculum for 1 h with rocking at 37 °C. Following infection, cells were washed twice with phosphate buffered saline (PBS; Gibco, Life Technologies, Carlsbad, CA, USA) and fresh media were added. Supernatants and cell lysates were collected over the course of time to determine viral titers, protein, and mRNA levels at each time point. Persistently infected cultures were obtained using previously described methods [52].

Viral titers were obtained by an immunofocus assay as previously described [54].

### 2.3. Western Blot Analyses

Cell lysates were obtained by pelleting trypsinized cells by centrifugation at 3000× *g* and 4 °C for 5 min, then lysing the pellets in ice-cold M-PER buffer containing phosphatase and protease inhibitors (Thermo Fisher, Waltham, MA, USA). Total protein was quantified using a bicinchoninic acid (BCA) assay (Thermo Fisher, Waltham, MA, USA). A total of 25 µg of the lysate was loaded into each lane on a 4–12% bis-tris gel and resolved using electrophoresis. Upon completion, protein was transferred onto a PVDF membrane using an iBlot system (Thermo Fisher, Waltham, MA, USA). The membranes were blocked for 45 min at RT in TBS-based Odyssey blocking buffer (Licor, Lincoln, NE, USA). The blocked membranes were incubated with primary antibodies at 1:1000 dilution in 1× tris buffered saline with Tween 20 (TBST) overnight at 4 °C, unless otherwise stated. Primary antibodies were purchased from Cell Signaling Technologies (Danvers, MA, USA) and these were: anti-AKT1 (C73H10), anti-AKT2 (D6G4), anti-AKT3 (4059), anti-phospho AKT S473 (D9E), anti-phospho AKT T308 (D25E6), anti-caspase-7 (9492), anti-cleaved caspase-7 (D6H1), anti-XIAP (D2Z8W), anti-survivin (71G4B7), and anti-beta-actin (8H10D10, 1:5000). Anti-phospho-PI3K from Thermo Fisher (Waltham, MA, USA) and anti-langat virus E 11H12 (a kind gift from Dr. Connie Schmaljohn, USAMRIID, Fort Detrick, Frederick, MD, USA) antibodies were also used.

Following incubation of the membranes with the primary antibodies, the membranes were washed three times in TBST and then incubated with IRDye secondary antibodies ((Licor, Lincoln, NE, USA) in TBST containing 0.015% SDS for 1 h at RT. The secondary antibodies were removed and followed by 3 more TBST washes. The membranes were scanned on an Odyssey CLx scanner ((Licor, Lincoln, NE, USA). Western blot bands were quantified using Image Lab software (Licor, Lincoln, NE, USA), and statistics were calculated in Graphpad Prism (Version 8, Graphpad Software, La Jolla, CA, USA).

### 2.4. Quantitative PCR (qPCR)

Cell lysates for qPCR were obtained by pelleting cells as described above, washing cells twice with PBS, and lysing cells in buffer RLT (Qiagen, Germantown, MD, USA). Genomic DNA was removed by running lysates through a RNeasy gDNA extractor column (Qiagen, Germantown, MD, USA). Total RNA was extracted using RNeasy kits (Qiagen, Germantown, MD, USA) and was quantified using a Nanodrop spectrophotometer (Thermo Fisher, Waltham, MA, USA). cDNA was synthesized from total purified RNA using the Superscript VILO kit (Thermo Fisher, Waltham, MA, USA). qPCR reactions were prepared with 1× TaqMan Fast Advanced master mix (Thermo Fisher, Waltham, MA, USA), 1× TaqMan assays, and cDNA and performed on a QuantStudio 6 real-time PCR machine (Thermo Fisher, Waltham, MA, USA). TaqMan assays used for qPCR were: AKT1 (assay ID Hs00178289), AKT2 (assay ID Hs01086099), AKT3 (assay ID Hs00987350), and ActB (assay ID Hs01060665_g1), and these were purchased from Thermo Fisher (Waltham, MA, USA).

### 2.5. CRISPR Knock Out of AKT Isoforms

AKT1 or AKT2 were knocked out using a CRISPR plasmid containing two gRNA (AKT1: CAGGUACUCAAACUCGUUCAUGG and GGAACGGCUUUCACGGGAACGGG; AKT2: CAGGCGGUCGUGGGUCUGGAAGG and GGGGGCAACCGUAUGAGGUAGUG), Cas9, and GFP (Atum, Newark, CA, USA). To knock the genes out, 293T cells were seeded in fibronectin-treated 6-well plates at 5 × 10^5^ cells per well and incubated overnight. The next day, 1 ug of the plasmid was transfected using Effectene (Qiagen, Germantown, MD, USA). GFP expression was confirmed using fluorescence microscopy, and cells were single-cell cloned and propagated in a stepwise fashion from 6-well plates until T75 flasks. Gene knockout was validated by western blot analyses.

### 2.6. Cell Counting Assay

Upon termination of the acute phase of infection, characterized by >95% cell death and the survival of a few cells [52], 293T cells were trypsinized and pelleted at 1500× *g* for 5 min at 4 °C. The pellets were resuspended in 1 mL complete media. An aliquot of the cellular stock was diluted 1:10 in 0.2% trypan blue and counted on a Countess cell counter (Invitrogen, Carlsbad, CA, USA).

### 2.7. Immunofluorescence Analyses

293T cells were seeded at 5 × 10^4^ in 4-well LabTek chamber slides (Thermo Scientific, Pittsburgh, PA, USA), which were pretreated with 10 µg/mL fibronectin (Sigma-Aldrich, St. Louis, MO, USA). The cells were either mock-infected or infected with LGTV TP21 at a MOI of 0.1. Upon termination of the acute phase of LGTV infection, the cells were fixed with 4% PFA and stained as described previously [52]. The fixed and stained cells were imaged on a Zeiss LSM 710 confocal microscope.

### 2.8. TUNEL Staining

293T cells were seeded at 6 × 10^4^ in 4-well LabTek dishes, which were pretreated with 10 µg/mL fibronectin. The cells were mock-infected or infected with virus as described in at a MOI of 1. Upon termination of the acute phase of LGTV infection, the cells were fixed with 4% paraformaldehyde (PFA) and stained as described previously [52]. TUNEL staining was performed using the in situ Cell Death Detection Kit (Sigma-Aldrich, St. Louis, MO, USA) following the manufacturer’s instructions. The fixed and stained cells were imaged on a Zeiss LSM 710 confocal microscope.

### 2.9. Monolayer Staining with Coomassie Blue

To visualize the cytopathic effect of LGTV on infected cell monolayers, infected cells were washed twice in PBS and fixed with 4% PFA for 10 min at room temperature. The PFA was aspirated followed by washing twice with PBS. Cells were stained with a Coomassie brilliant blue stain (G-250; Thermo Fisher, Waltham, MA, USA) for 5 min. The stain was removed, and cells were washed twice with PBS and imaged using an AxioVert.A1 microscope equipped with a Zeiss Axiocam 503 monochromatic camera.

## 3. Results

TBFVs utilize host machinery to replicate and persist, and part of this hostile take-over can involve changes in gene expression [50]. We previously demonstrated that LGTV causes transcriptomic changes during both the acute and persistent phases of infection. *AKT2* mRNA was shown to be upregulated in the persistently infected cells compared to acutely infected cells, suggesting that the PI3K-AKT pathway is utilized for preventing apoptosis during persistence [50]. A clearer understanding of this pathway’s involvement in the establishment and maintenance of TBFV persistence might illuminate new drug targets to combat TBFV persistence.

We began by dissecting the AKT response during LGTV infection. Using phospho-pan-AKT antibodies, we observed that phosphorylated AKT (pAKT), the active form of AKT, decreased over time in the acute phase of infection (Figure 1a). This decrease was seen at both canonical phosphorylation sites, T308 and S473. Interestingly, pAKT S473 began decreasing in infected cells at 72 hpi, and T308 began decreasing at 96 hpi. This finding suggested that the decrease in pAKT was controlled by a change in AKT isoform expression, rather than by a change in kinase or phosphatase activity [40]. The virus-induced effects were more pronounced during the later stages of infection at 96–120 hpi, which was associated with higher levels of intracellular viral E protein (Figure 1a).

A decrease in AKT expression at 48 hpi at an MOI of 5 in DENV and JEV infections was also observed, but this was not associated with a relative decrease in pAKT levels [36]. More importantly, these MBFVs caused an increase in pAKT levels within 30 min post infection. We did not observe a similar early increase in pAKT, but the authors in the MBFV study studied lysates from 5 min to 48 h post infection, used a MOI of 5, and infected with MBFVs, which might explain the discrepancy.

To explain why pAKT decreases during infection, we examined the relative levels of each AKT isoform. Surprisingly, viral infection differentially modulated the expression of each isoform (Figure 1a). AKT1 and AKT2 levels decreased as infection progressed, compared to the mock infected cells. In contrast, AKT3 expression was substantially increased by 96 hpi (Figure 1a) in comparison to the mock infected cells. Mock infected cells showed a gradual decrease in AKT3 expression throughout the time course. This decrease in AKT3 in mock infected cells may be due to these cells reaching confluency and attempting to slow growth by reducing levels of pro-survival AKT3. The increase in AKT3 in infected cells may be a compensatory increase due to decreasing AKT1 and AKT2 levels. However, the higher levels of AKT3 were not associated with higher levels of total pAKT, suggesting that AKT3 was less phosphorylated at T308/S473 compared to the other two isoforms. This disconnect could be due to a lower kinase affinity for AKT3 or an antibody detection effect. AKT3 is upregulated in malignant melanomas and is uniquely necessary among AKT isoforms for cranial development [49,55]. As HEK 293T cells are not neuronal, AKT3 may not be exerting its full effect in this cell line.

In support of the protein level data, the mRNA expression of *AKT1* decreased during acute LGTV infection decreased, although not significantly, while *AKT2* and *AKT3* mRNA levels remained constant when compared to respective mock levels (Figure 1b).

To confirm whether pAKT levels were controlled via AKT1/2 expression, we investigated alternative mechanisms that control pAKT levels. AKT phosphorylation is modulated by PIP3, whose concentration is positively regulated by PI3K and negatively regulated by PTEN. PI3K is activated upon phosphorylation. To confirm that the decrease in pAKT is controlled by reduction in AKT expression and not PIP3 machinery, we investigated the relative expression levels of PTEN and pPI3K by Western blot analysis (Figure 2a). We discovered that there was no obvious difference between infected and mock levels of either PTEN or pPI3K; thus, AKT phosphorylation was controlled by factors other than PIP3 concentration.

To examine a role for caspases in the decreased AKT levels, we examined the activation of effector caspases 3 and 7 during infection and observed higher levels of cleaved caspase 3 in infected cells during infection at 120 hpi, but no difference in cleaved caspase 7 between the uninfected and infected cells (Figure 2b). This demonstrated that effector caspases were activated during TBFV infection but were only detectable after AKT1 and AKT2 depletion had begun.

We also investigated whether AKT1 or AKT2 were individually required for LGTV infection in HEK 293T cells. Using CRISPR technology, we developed HEK 293T cells lacking either AKT1 or AKT2 (Figure 3a). We hypothesized that AKT knockouts would not enter viral persistence, or at least undergo a massive cytopathic effect (CPE) event, like the WT cells. Surprisingly, we found that AKT1 KO and AKT2 KO cells survived the acute phase of infection in greater numbers compared to the WT cells (Figure 3b,c). Six times as many AKT1 KO, and eight times as many AKT2 KO cells survived the acute phase as WT cells (Figure 3c).

Although AKT is regarded as a cornerstone of cell survival and we anticipated that cells deficient in an AKT isoform would be crippled in terms of cell survival, higher numbers of AKT1 KO and AKT2 KO cells survived acute LGTV infection than WT cells. To explain this phenomenon, we investigated the relative levels of AKT in the three cell lines at the beginning of the LGTV-induced lytic crisis. AKT1 KO and AKT2 KO cells at 120 hpi had substantially higher amounts of pAKT S473 than WT cells, and AKT2 KO cells had significantly higher amounts of pAKT T308 than WT cells (Figure 4a).

XIAP and survivin are upregulated by pAKT to inhibit apoptosis, and XIAP is a potent inhibitor of caspase cleavage and may form a complex with survivin [56]. Survivin is reported to have pleiotropic roles in mediating not only apoptosis, but it also has a role in mitosis [56]. We investigated if the increase in pAKT was associated with higher levels of XIAP and survivin. Densitometry measurements of pAKT S473 and T308 and survivin showed a consistently higher levels of these proteins and significantly higher levels of XIAP in AKT1 KO and AKT2 KO cells (Figure 4b). In addition, LGTV-infected AKT1 KO and AKT2 KO cells had higher levels of either AKT2 or AKT1, respectively, than WT cells, further suggesting that depletion of one of these isoforms causes a decrease in pAKT during infection (Figure 4c). We confirmed that higher levels of pAKT, XIAP, and survivin assist AKT1 KO and AKT2 KO in resisting LGTV-induced cell death by TUNEL staining at 168 hpi (Figure 4d). Almost all the WT cells at 168 hpi were TUNEL positive, while a greater number of the AKT1 KO and AKT2 KO cells were TUNEL negative. These results led us to conclude that the AKT1 KO and AKT2 KO cells survive LGTV-induced cell death due to an intact AKT signaling system and demonstrate that LGTV-induced death in 293T cells requires both AKT1 and AKT2.

Next, to investigate whether the KO affected virus replication and whether virus replication affected pAKT levels, we titrated the virus during the acute phase of infection and the start of the viral persistence, termed PST0. During most of the acute phase (24–120 hpi) and the start of persistence, AKT1 KO and AKT2 KO cells shed virtually the same amount of infective virus as the WT cells (Figure 5a). However, after 120 hpi, the virus titer in the AKT1 KO and AKT2 KO cells was higher than the WT virus titer, reaching a 7-fold difference at 168 hpi. In all three cell lines, virus titers increased to 10^7^ FFU/mL by the start of persistence (Figure 5a). To confirm that at the end of acute phase all the surviving cells in the AKT1 KO and AKT2 KO cultures were infected, we performed immunofluorescence staining for the viral E protein. We saw that almost every cell in the three cell lines was infected (Figure 5b). Our results here demonstrated that AKT1 KO and AKT2 KO cells received the same viral burden as the WT cells. These results do not explain why the AKT1 KO and AKT2 KO cells had higher levels of pAKT at 120 hpi and survived acute LGTV infection better.

## 4. Discussion

We have previously demonstrated that HEK 293T cells infected with LGTV undergo a lytic crisis, which kills most of the cells, but a few survive to become persistently infected. The evasion of apoptosis by a few cells has been intriguing. Transcriptomic work suggested an upregulation of mRNA encoding for AKT2 [50]. Therefore, we set out to investigate changes in AKT expression levels during the acute phase of LGTV infection. We observed that AKT1 and AKT2 levels decrease over time as infection progressed, but AKT3 was dramatically increased at 120 hpi.

A decrease in AKT expression in DENV and JEV infections was reported, but this was not associated with a relative decrease in pAKT levels [36]. More importantly, these mosquito-borne flaviviruses caused an increase in pAKT levels within 30 min post infection. We did not observe a similar early increase in pAKT, and differences in the MOI used to conduct the experiments may explain these differences. Furthermore, the early activation of AKT in the MBFVs was to evade early apoptosis [36].

Cells lacking either AKT1 or AKT2 surprisingly survived LGTV-induced cell death better than WT cells (Figure 3). While AKT isoforms arise from separate genes, many of their functions overlap. However, our data suggest that AKT1 and AKT2 have nonredundant functions and may also be in involved in crosstalk. Other studies have shown that each AKT isoform causes expression of different microRNAs, which can affect the activation of the other isoforms [57,58]. In one study, AKT2 was shown to induce expression of miR-21 during hypoxia, which causes a downregulation of PTEN and an upregulation of active AKT1 and AKT2. In addition, another study showed that knockdown of AKT1 in human lens epithelial cells causes higher expression of AKT2, higher levels of pAKT, and greater cell survival during oxidative stress [57]. These cells, however, were more susceptible to oxidative-stress-induced apoptosis after knocking down AKT2 individually, and simultaneously knocking down AKT1 and AKT2 [59]. In contrast to these results, another study found that AKT1/2 double KO MEFs are less susceptible to oxidative stress than WT cells, suggesting a diverse AKT response to different stressors [60]. In our system, AKT1 and AKT2 may regulate different processes and protect or sensitize cells to different stressors. Additionally, this can explain why AKT1 KO cells die more than AKT2 KO cells. Using proteomic and transcriptomic analyses to compare the cellular environments of WT, AKT1 KO, and AKT2 KO cells might provide new insights on the individual contributions of these proteins to LGTV-induced cell death and their intertwined signaling networks. We also attempted to make AKT1/2 double KO 293Ts but were unsuccessful.

Because most of the cells infected during the acute phase of LGTV will die, we deduce that that outcome is because ATK1 or AKT2 levels decrease and AKT3 levels increase dramatically. However, in the background of these differential changes, there are cells that sense decreased expression levels of the AKT isoforms. These cells may then be able to further upregulate the unaffected isoform and in turn upregulate XIAP and survivin to escape the lytic crisis and become persistently infected. However, additional studies are required to prove this hypothesis.

AKT is primarily considered a cell survival protein. Its functions can modulate different aspects of the apoptosis pathway including TRAIL expression, BAD phosphorylation, XIAP stabilization, and survivin expression [61,62]. Our results here confirm that pAKT, XIAP, and survivin are functionally linked, in that when the isoforms were individually depleted, there was an upregulation of XIAP or survivin (Figure 4a). Downregulation of pAKT was associated with a downregulation of XIAP and surviving during LGTV infection, and an upregulation of pAKT is associated with an upregulation of XIAP and survivin. XIAP and survivin are known inhibitors of apoptosis (IAP). XIAP is a potent inhibitor of caspase cleavage and may form a complex with survivin, which stabilizes XIAP [56]. These IAP proteins were associated with a protection from LGTV-induced apoptotic cell death, as demonstrated with TUNEL staining (Figure 4a). These results suggest a defined pathway for how loss of pAKT can sensitize cells to LGTV-induced apoptotic cell death.

In summary, we have shown that pAKT decreased during acute flavivirus infection. This was associated with a decrease in AKT1 and AKT2 levels and an increase in AKT3 protein levels. There was no obvious change in PTEN or pPI3K levels during the same time-points, suggesting that pAKT decreased because of lower AKT1/2 expression. Loss of pAKT at this late stage will have profound effects on the many cell processes AKT controls.

We also discovered that cells individually lacking AKT1 or AKT2 survive LGTV infection better than WT cells, and this was associated with higher amounts of antiapoptotic XIAP and survivin and increased resistance to LGTV-induced apoptotic cell death. Further work using AKT3 KO cells, multiple isoform KO cells, and whole cell proteomic and transcriptomic analyses can illuminate more steps in this fascinating pathway.

## Figures and Tables

**Figure 1 viruses-12-01059-f001:**
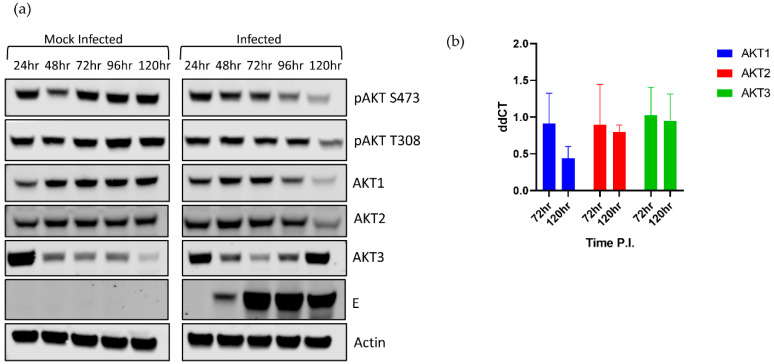
Langat virus (LGTV) causes differential expression levels of the AKT isoforms. 293T cells were infected with LGTV at a multiplicity of infection (MOI) of 0.1 and harvested at the indicated timepoints. (**a**) Representative Western blots showing AKT isoform levels as well as phosphorylation patterns. (**b**) mRNA analysis of *AKT* isoforms in infected and mock infected cells. AKT isoform mRNA was evaluated compared at 72 and 120 hpi. The observed differences were not statistically significant.

**Figure 2 viruses-12-01059-f002:**
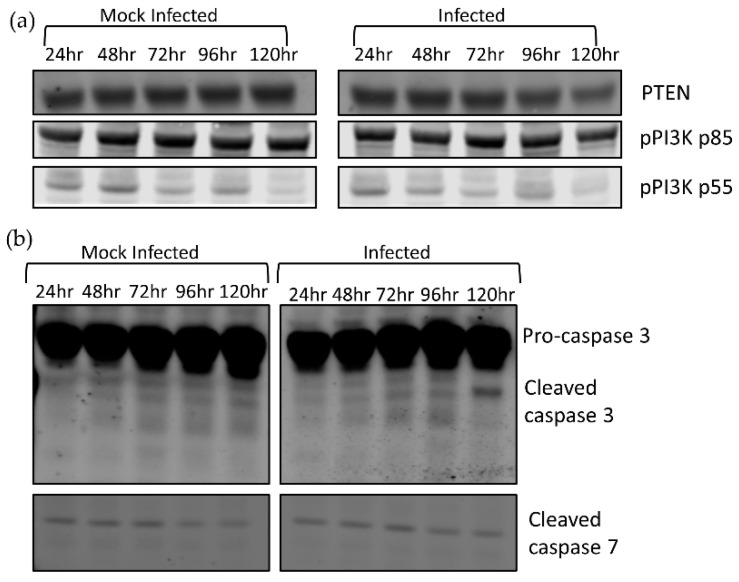
Analysis of factors responsible for AKT phosphorylation. 293T cells were infected with LGTV at an MOI of 0.1 and harvested at the indicated timepoints. (**a**) Western blot detecting PTEN, pPI3K, and pPI3Kp55 levels; (**b**) detection of cleaved caspase 3 cleavage at 120 h post-LGTV infection.

**Figure 3 viruses-12-01059-f003:**
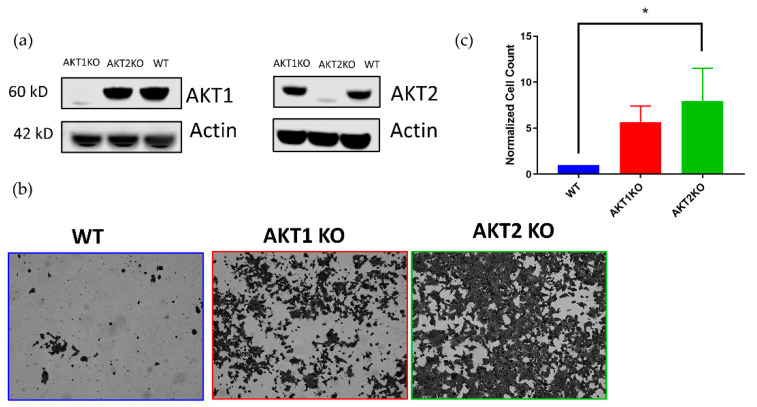
AKT1 knockout (KO)and AKT2 KO 293T cell lines resist LGTV-induced a cytopathic effect (CPE). (**a**) Validation of CRISPR-mediated AKT1 and AKT2 knockouts in 293T cell lines with Western blotting analysis. (**b**) Stained monolayers comparing the susceptibility of AKT knock out cells to apoptosis. AKT1 KO, AKT2 KO, and WT 293T cells were infected with LGTV at an MOI of 0.1 and imaged at 168 hpi. Images were taken with a 10× objective. (**c**) Quantification of surviving cells at 168 hpi. * *p* < 0.05 (one-way ANOVA followed by Tukey’s multiple comparison test). Error bars indicate standard deviation from four independent experiments.

**Figure 4 viruses-12-01059-f004:**
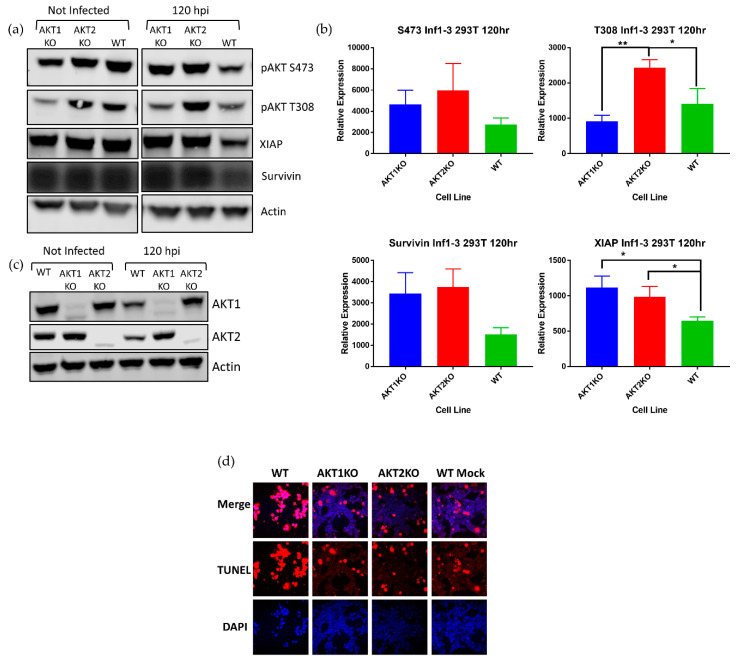
Higher pAKT levels in AKT KO cell lines show resistance to LGTV-induced apoptosis. AKT1 KO, AKT2 KO, and WT 293T cells were infected with LGTV at an MOI of 0.1. (**a**) Comparison of pAKT, XIAP, and survivin levels in wild-type and AKT knock out cells at 120 hpi. (**b**) Densitometry measurements of the blots in A show that there is significantly more XIAP present in AKT1 KO and AKT2 KO cells than WT cells at 120 hpi, * *p* < 0.05, ** *p* < 0.01 (one-way ANOVA followed by Tukey’s multiple comparison test). Error bars indicate standard deviation from three independent experiments. (**c**) In addition to higher levels of pAKT, AKT1 KO and AKT2 KO cells have more AKT1 or AKT2. (**d**) TUNEL staining (red) at 168 hpi shows substantially more apoptosis in WT cells than AKT1 KO and AKT2 KO cells. Nuclei were counterstained with DAPI (4′,6-diamidino-2-phenylindole; blue). Images were taken with a 40× objective.

**Figure 5 viruses-12-01059-f005:**
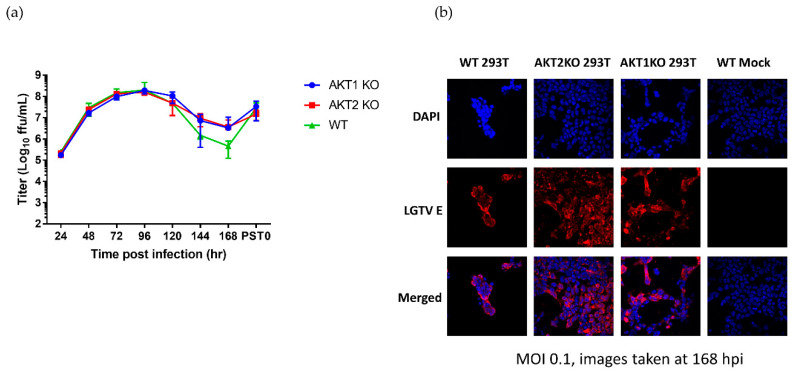
LGTV-infected AKT knockout cells enter viral persistence without substantial alteration in viral replication. AKT1 KO, AKT2 KO, and WT 293T cells were infected with LGTV at an MOI of 0.1. (**a**) Viral titers of acute and persistently infected AKT KO and WT cell lines. Error bars indicate standard deviation from three independent experiments. (**b**) Immunofluorescence analysis evaluating expression of the viral E protein (red), a proxy for infection, at 168 hpi. Nuclei were counterstained with DAPI (blue). Images were taken with a 40× objective.

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
