# Peer review of "Tick-Borne Flaviviruses Depress AKT Activity during Acute Infection by Modulating AKT1/2"

_viruses, 2020, doi:10.3390/v12101059_

Round 1

Reviewer 1 Report

This manuscript by Kirsch et al. describes the significance of changes in AKT level during Langat virus infection. The study is succinct and performed with proper controls. Here, results are convincing and presenting a significant but understudied area of virus persistence. The study is a continuation of previously published work on AKT. The authors also present clear evidence for a high level of AKT during persistent infection. Authors have characterized the role of individual viral proteins in inducing the AKT level and found the NS4B and M proteins are capable of inducing its expression. The authors also generated CRISPR knockout cells for AKT1 and AKT2 and lentivirus-driven overexpression of AKTs. Finally, the authors conclude that the depletion of AKT1 and AKT2 causes Langat virus-induced cell death.

Comments

Although the manuscript describes a specific phenomenon, the writing gets complicated in between with a large number of AKTs and its different levels of expression. Authors may benefit from a graphical representation of the role of AKT in Langat virus infection.

The lentivirus data is not convincing, as written. It may be removed, or additional experiments may be needed. The explanation of results from the overexpression is not clear.

17: active AKT (phosphorylated AKT): this has been used inconsistently

85: G-coupled protein receptors 

124: ‘Fed’ may be changed to ‘add’

134: DNA plasmid may be changed to plasmid DNA

157: Please list primer sequences

179: ‘The day before transfection, 2 x 104 cells into each well of a 24-well plate.’: rewrite

201: describe ‘inoculum media and 1 mL incubation media’. 

209: Specify which Coomassie blue 

317: It will be better if western blot analysis of viral proteins is included in this figure. For instance, since M and E are dimers, how stable is the separate expression of M and E?

358 and 359: Introduce survivin and XIAP

397: Fix ‘Immunofluorescent

403: Provide a western blot for the restoration of AKT expression

      414: ‘data suggests’ 

Reviewer 2 Report

This manuscript describes a study investigating the association of differential levels of the three different AKT isoforms and phosphorylated AKT (2 sites) in modulating the death of HEK293T cells acutely infected with the tick-borne virus Langat (LGTV) and in the survival of persistently infected cells. The involvement of the AKT/PI3K pathway in the survival of cells infected with TBEV and also of cells infected with several mosquito-borne flaviviruses has been previously reported in multiple studies but the levels of individual AKT isoforms were not analyzed. It was previously shown that upregulation/activation of the pro-survival AKT/PI3K pathway occurred at very early times after infection and suppressed early Caspase 3 cleavage promoteing infected cell survival until later times of infection when the activation of the AKT/PI3K pathway is no longer sufficient to prevent apoptosis.

The stated goal of this study is to investigate the involvement of the PI3K-AKT pathway in tick-borne flavivirus persistence. A previous transcriptome study by the authors showing that AKT2 mRNA was upregulated in LGTV persistently infected Hek293T cells compared to acutely infected cells and a few additional studies on other viruses indicating the involvement of components of this pathway in virus persistence were the impetus for initiating the current study. The levels of individual AKT isoforms and 2 pAKT species were analyzed at different times during acute infection and some were analyzed in persistently infected cultures. mRNA levels were analyzed at two later times of acute infection. Although the levels of two of the AKT isoforms were decreased at later times of acute infection, PTEN and pPI3K levels were not increased and a small amount of cleaved caspase 3 was observed at 120 hpi. The effect on the levels of two AKT isoforms and one species of pAKT of overexpression of individual viral proteins was next analyzed. AKT1 and AKT2 CRISPR KO cell lines were generated and tested for AKT levels, survival protein levels, cell death and virus production. The loss of one AKT isoform increased the level of another at 120 hpi and decreased the number of cells with a detectable TUNNEL signal. The slight decrease in cell death either of the infected KO cells was associated with slightly higher virus yields at later times after infection compared to WT and the slight decrease in cell number could be reversed by overexpression of AKT1 but not AKT2.

There are several concerns about the significance of the data. Although the goal of the study was to analyze the role of the AKT-PI3K pathway in persistence, the majority of the data presented were obtained from acutely infected cells. Although some variation in the levels of the three different AKT isoforms was observed in late stage infected cells, no clear correlation of these levels with the phenotypes was found.  Due to the relatively small proportion of the WT infected cells that die at late times after infection (Fig. 7), it would be expected that there would be a very high background of non-apoptotic infected cells contributing to the Western data at late times of infection. The data on the proportion of cells killed by the virus at later times of the acute infection is not consistent between Figs. 4 and 7. Western data are not shown to confirm that the overexpression of the rescuing isoform in the knockout cells (Fig. 7). Although AKT2 mRNA was previously observed by the authors to be increased in persistently infected cells, no data on AKT2 protein levels in persistently infected cells were included in this manuscript (Fig. 1C or Fig. 3). Another concern is a high degree of variation in the mRNA data shown in Fig. 1B and the omission of mRNA data for persistently infected cells. The value of the data showing differential effects of ectopic expression of single viral proteins on AKT isoform levels and phosphorylation to the study conclusions is questionable since the conditions in these cells are very different from those in late acute stage infected cells. Also, the validity of the data in this experiment is not supported due to the high level of experimental variation observed and the lack of data on the levels of the individual viral proteins expressed. Again, no data on AKT2 were included for this experiment. Knockout of AKT2 appears to increase acutely infected cell survival and yet previously the authors found that AKT2 mRNA was upregulated in persistently infected cells.  

The manuscript contains statements that over interpret the data. For instance, the authors state that the observed difference in the timing after infection of phosphorylation at AKT T308 and S473 is “controlled by a change in AKT isoform expression rather than by a change in kinase or phosphatase activity” and that their “results suggest a defined pathway for how loss of pAKT can sensitize cells to LGTV-induced apoptotic cell death.” The data presented do not support either of these conclusions.

The levels of the AKT and other pathway components are high in mock infected cells, some of them decrease at late times of infection and then again increase in persistently infected cells as would be expected. The data do not analyze what determines the shift from decreased levels to normal levels to allow continued survival of the persistently infected cells. No information about whether or not there is differential phosphorylation of individual isoforms is provided. The data suggest that the levels of individual AKT isoforms are not that important and loss of one isoform can be compensated by an increase in others. The level of pAKT phosphorylation at S473 seems to be important for keeping cells from going into apoptosis. However, the small number of cells that develop a “death phenotype” is low in this system making it difficult to identify critical factors. The overall conclusion that the AKT pathway is involved in cell survival does not differ from those of previous studies.

Editorial concerns

The manuscript is not well written. The Abstract does not clearly state what was found and why the data are novel or important. It is not stated in the Introduction that the AKT isoforms are produced from separate genes. The data obtained from previous studies about the involvement of the AKT-PI2K pathway in various types of virus infected cells is not clearly organized in a logical sequence that addresses the relevant aspects for the authors’ study (L67-82 and 97-103).

Many of the figure legends do not contain any experimental details for the experiments for which data are shown. Descriptions of the data obtained and conclusions drawn are inappropriately included in the figure legends instead of in the Results text. The meaning of some sentences is not clear. Statements about data are sometimes inaccurate and data are sometimes overinterpreted. 

L19- The use of the word AKT here is not clear since individual isoform levels were indicated for acutely infected cells.

L122- The source of the virus and how the stock virus was produced were not indicated.

The conclusions on L24-27 and L113-116 are not consistent with the data or with each other.

L178- Additional information is needed about the generation of the lentiviruses. Were the cell lines generated single cell cloned or bulk cultures? If bulk cultures, what % of these cells expressed the protein of interest?

L179- This is not a sentence as written.

L188- How was the termination of the acute phase of infection defined?

Fig. 1- It is not explained what time post infection is PST-0.

L201- The components of “inoculum media” and “incubation media” need to be defined.

L240- MBFV is not spelled out prior to abbreviation. L438- IAP needs to be spelled out before it is abbreviated.

L243- It is not clear from this statement whether any times before 24 hpi were tested or not. The effect was previously observed for other flavivirus infections at very early times after infection (at 3 hr or before). Also, the relative confluency of the cells could affect the results.

L257, 260, 286, 415, 442, 443, 444 - In each of these sentences, the word “expression” should be omitted since levels were tested.

L296-298- These sentences should be omitted.

L269, 273, 279-280, 309-310, 343, 360, 372, 414, 433 – these statements are not consistent with ether previous statements or the data shown.

Fig. 4B- It is not clear whether only very small numbers of cells were counted or if the legend is missing log units.

Fig. 5B- None of the graph legends can be read because the font is too small and the print is fuzzy. The Survivin bands cannot be distinguished well due to the very high background and is very broad. It is not clear that this is a specific band.

The Results section contains too much discussion. This information should be moved to the Discussion section.

L327- It is not clear which studies are being referred to here.

L381- …we titered virus yields produced during the acute phase   L383- …the same amount of infective virus … change to infectious virus

L402- The expression of the AKT2 lentivirus should have been tested by Western blot to validate the data.  

L432, 434- Surviving should be survivin.  This word was capitalized earlier in the manuscript.

L447- Loss of pAKT. A decrease not a loss was observed.

Reviewer 3 Report

The research article entitled “Tick-borne flaviviruses depress AKT activity during acute infection by modulating AKT1/2” by Kirsch et al extended their previous work and found that LGTV could hijack the AKT pathway for viral persistent infection. In general, the data are very interesting and significant to the field. I have some minor concerns.

  1. Figure 1(a): I would like to hear from the authors why the AKT3 dramatically reduced after 24 hours in both mock and infected groups. Is it caused by cell deaths (likely not because the Figure 2(b) shows no differences in Cas 3 and 7 between the mock and the infected groups)? If so, can the authors perform the experiment with a lower confluent cell?
  2. Figure 2: the viral protein blot needs to be shown.
  3. Figure 3(a): the viral protein productions need to be shown.
  4. Some typos can be seen, please check thoroughly.

Round 2

Reviewer 2 Report

The manuscript has been improved by the deletions and additions made by the author in response the the reviewer comments. There are a few remaining concerns.

L14- The word infection is too vague here.  …its role  “during the virus infection cycle”   or “in infected cells”

L26- Not a sentence as written- could combine two sentences   …wild-type cells due to higher amounts….

L95-96- Please add info about these 3 isoforms being produced by different genes here unless this is not true and they are splice variants. The requested information was added than deleted on L113-114.

L244-245- The cells and MOI and time of harvest used for each experiment should be stated in the Fig legend or Results text. The information was deleted in two of the revised legends and should not have been.

L254-255- This statement does not seem to be consistent with the one on L275-277.  

L296-299- The data in Fig. 1 do not support these statements.

L275- …data, the AKT1 mRNA levels decreased during the acute LGTV infection, while …

L364- The data in Fig, 3C does not appear to show the same 8-fold change for both AKT1 KO and AKT2 KO cells.

L383- …if the increase…

L412-413- This sentence does not include the other proteins analyzed.

L461- MBFVs should be spelled out. The context for the information described in this statement is not clear as written.

L483- …will die…
